# A Systematic Review on the Potential of Aspirin to Reduce Cardiovascular Risk in Schizophrenia

**DOI:** 10.3390/brainsci13020368

**Published:** 2023-02-20

**Authors:** Joseph Dao, Savreen Saran, Melody Wang, Christina Michael, Nhu-y Phan, Alfredo Bellon

**Affiliations:** 1College of Medicine, The Pennsylvania State University, Hershey, PA 17033, USA; 2Department of Psychiatry and Behavioral Health, Penn State Milton S. Hershey Medical Center, Hershey, PA 17033, USA

**Keywords:** schizophrenia, aspirin, cardiovascular, anti-inflammatory, mental-health, metabolic

## Abstract

Cardiovascular disease (CVD), including heart disease and stroke, continues to be the leading cause of death worldwide. Patients with mental health disorders, including schizophrenia (SCZ) are known to have an increased risk for CVD. Given the association with metabolic syndrome, patients with SCZ are often prescribed metformin and statins but its impact remains unsatisfactory. The use of aspirin (ASA) to decrease cardiovascular risk in the general population has been thoroughly investigated and clear guidelines are currently in place. Since adjuvant treatment with ASA could possibly decrease CVD risk and mortality in SCZ, we conducted a systematic review of the literature to determine the state of the current literature on this subject. Our systematic review points to gaps in the literature on CVD prevention in SCZ and illustrates an obvious need for further research. Although several studies have shown increased CVD risk in SCZ, to date, no research has been conducted on the utilization of CVD preventative treatment such as ASA for SCZ.

## 1. Introduction

Cardiovascular disease (CVD), including heart disease and stroke, continues to be the leading cause of death worldwide. Between 1990 to 2019, prevalence of CVD cases nearly doubled from 271 million to 523 million, and the number of CVD deaths increased from 12.1 million to 18.6 million [1]. There are known patient populations that have an increased risk for CVD, including patients with mental health disorders such as schizophrenia (SCZ) [2]. It has been previously established that the life expectancy of patients with SCZ is significantly shorter in comparison to the general population [3,4] and that the leading cause of death is CVD [2,5,6,7,8]. This increased risk is compounded by several other characteristics associated with SCZ, such as sedentary lifestyle, high rates of smoking, and diets rich in calories and fat [9]. Furthermore, patients with SCZ are treated with antipsychotics, for which side effects include; obesity, dyslipidemias, glucose intolerance, hyperinsulinemia, and hypertension, all of them highly predictive of CVD [10].

Given the association of metabolic syndrome with SCZ, these patients are often prescribed metformin and statins [11,12,13] but the impact on life expectancy remains limited [4]. The use of selective serotonin reuptake inhibitors (SSRIs) has been proposed as a potential treatment alternative but this option remains under investigation [3]. In the general population, adults with increased CVD risk are often prescribed aspirin (ASA). Current U.S. Preventive Services Task Force (USPSTF) guidelines recommend the use of ASA in primary prevention of CVD in adults 40–59 who have a 10% or greater 10-year CVD risk. More recent studies, however, have recommended against using ASA as an intervention for decreasing CVD risk due to increased risks of bleeding outweighing the benefits. Three studies well known in the literature are ARRIVE, ASPREE, and ASCEND. ARRIVE evaluated the ASA’s ability to reduce risk of initial vascular events. After following 12,546 enrollees for 72 months, the results indicated there was no change in preventing cardiovascular events within the ASA group compared to placebo [14]. ASPREE focused on the effects of low-dose ASA compared with placebo among healthy individuals. With 19,114 enrollees followed for 4.7 years, ASA did not improve disability-free survival or improve major adverse cardiovascular events [15]. ASCEND assessed whether ASA was effective for patients with diabetes and no history of CVD. It followed 15,480 patients for 7.4 years, and found an improvement in major cardiovascular events in the ASA group compared with placebo [16]. The ongoing research and availability of literature on ASA utilization allows for constant refinement of USPSTF guidelines for the general population. While guidelines for ASA use outlining primary and secondary prevention of CVD exist for the general population, to our knowledge there are no guidelines for ASA use in SCZ, even though these patients have a high risk for CVD. An example of a population for which ASA is useful in preventing CVD is that of patients with diabetes. Even though not all patients with diabetes benefit from ASA, there are clear clinical instructions on when and how to prescribe it [17].

In this manuscript we first present evidence on the current understanding of how patients with SCZ are at an increased risk for CVD. Secondly, we conducted a systematic review of the literature to determine if there is evidence to support the use of ASA in patients with SCZ to help curtail their cardiovascular risk. Lastly, we discuss the potential of ASA use in patients with SCZ to mitigate CVD risk.

## 2. Materials and Methods

A systematic literature search was conducted from the earliest publication date available through 5 January 2023, in accordance with the Preferred Reporting Items for Systematic Reviews and Meta-analyses (PRISMA) 2020 guidelines.

### 2.1. Data Sources and Search Strategy

Independent investigators searched PubMed, PsycINFO, Cochrane Library, Web of Science, and Scopus for published articles. The search was limited to studies published in English. Keywords and MeSH heading picked based on Patient or Population, Intervention, Control or Comparison, and Outcome (PICO) search strategy and included “schizophrenia”, “antipsychotic agents”, “aspirin”, “anti-inflammatory agents, non-steroidal”, “neuroleptic agents”, “antipsychotic drug”, “neuroleptic drug”, “neuroleptics”, “antipsychotics”, “anti-psychotics” “cardiovascular diseases”, “cardiovascular risk”, and “cardiovascular syndrome, metabolic”.

### 2.2. Study Selection

The study selection process is illustrated in Figure 1. All references were collated on Endnote Web. After removal of duplicates using the function on Endnote Web, the remaining articles were subjected to a screening and review. In the screening stage, papers were excluded based on their title and abstract if they did not clearly report on patients with SCZ spectrum disorder or ASA use. To reduce the chance of omitting relevant articles, reviewers were instructed to be overly inclusive. Papers were retained if there was insufficient information to exclude them. This was performed by four independent investigators. The remaining articles were reviewed. The purpose at this stage was to more closely assess studies based on the inclusion and exclusion criteria. Where necessary, full text was reviewed. Complete inclusion and exclusion criteria included: (1) peer- reviewed publication, (2) in English language, (3) includes patient(s) with SCZ spectrum disorder, (4) includes discussion or data about using ASA as a treatment to decrease CVD risks.

## 3. Results

After removal of duplicates, a total of 484 articles were identified and screened. A total of 429 articles were excluded during screening based on their title and abstract not clearly reporting on patients with SCZ spectrum disorder or ASA use. The remaining 55 articles included clinical trials, literature reviews, meta-analyses, opinions, and observational studies. These articles were reviewed by three independent investigators in parallel and independently. All 55 articles were excluded after applying the complete inclusion and exclusion criteria. Nonetheless, there is significant evidence indicating patients with SCZ have an increased risk of CVD due to a variety of factors that we present in the following paragraphs. We also consider that one of the manuscripts yielded by our systematic search is worth presenting in our results. Even though it does not meet our full inclusion criteria, it is the only manuscript published to date assessing the use of ASA in SCZ for purposes related to CVD risk.

### 3.1. Schizophrenia and Cardiovascular Risk

There is considerable evidence indicating patients with SCZ have an increased predisposition for CVD. For instance, drug naive patients were found to have impaired fasting glucose tolerance, with significantly higher fasting plasma levels of glucose, insulin, and cortisol levels [18]. Body fat distribution is also a problem. Patients with SCZ carry three times as much intra-abdominal fat than age- and BMI-matched control subjects [19]. This type of fat distribution correlates with increased insulin resistance [19]. Other characteristics of SCZ that increase CV risk include experiencing apathy and anhedonia, subsequently leading to a more sedentary lifestyle [20]. Two other lifestyle factors that also increase CV risk in SCZ are smoking and diet. Smoking prevalence is higher in patients with SCZ when compared to the general population [21]. In fact, it is estimated that more than 60% of SCZ patients are currently smokers [21]. The combination of these behaviors, in addition to the diets of patients with SCZ commonly found to be rich in calories and fat ultimately results in development of dyslipidemias, insulin resistance, diabetes mellitus type 2, and CVD [9,22]. Moreover, such conditions are not adequately treated. The rates of untreated dyslipidemia and hypertension in SCZ have been reported to be up to 88% and 62%, respectively [23]. Direct management of CVD is also deficient as patients with SCZ are 47% less likely to receive cardiac intervention than those without mental illness [23]. Not surprisingly, individuals with SCZ and ischemic disease have a shortened life expectancy by at least 10 years [24,25]. However, even with appropriate management, outcomes in SCZ are not always as expected. A recent meta-analysis indicated that Acute Coronary Syndrome in patients with SCZ was associated with a significantly higher risk of mortality after one month and during a follow-up of greater than one year, both with and without the adjustment of revascularization treatment [26]. For patients who received cardiac revascularization procedures, those with co-existing severe mental illness (SMI) were found to have higher risks of major adverse cardiac events such as death, myocardial infarction, and target vessel revascularization [27]. Furthermore, the study also found that patients with SMI are less likely to receive recommended medical treatment after percutaneous coronary intervention [27].

Another factor to consider for the high prevalence of diabetes and CVD risk is antipsychotic intake. Both first-generation antipsychotics (FGA) and second-generation antipsychotics (SGA) have been known to cause metabolic abnormalities [10]. The rate at which these side effects occur is what differs among these medications. For FGAs or typical antipsychotics, lower potency antipsychotics such as chlorpromazine and thioridazine were found to induce greater weight gain than their higher-potency counterparts fluphenazine and haloperidol [19]. However, SGAs or atypical antipsychotics are still more commonly associated with metabolic side effects, especially with olanzapine and clozapine. With clozapine, weight gain lasted for as long as 30 weeks after initiation of treatment, with some patients gaining >30% of their total body weight [19]. Olanzapine also had significant weight gain and can persist for up to 1 year [19]. Weight gain for risperidone and quetiapine appears to be intermediate and associated with dose [19]. Though the mechanism of weight gain via antipsychotic use is currently undetermined, its side effects should remain in consideration when analyzing CVD risk in individuals with SCZ. The factors that increase CVD risk in SCZ patients along with potential treatment approaches to decrease this risk is demonstrated in Figure 2.

### 3.2. Emerging Data on the Use of Aspirin in Patients with Schizophrenia

Currently, mitigation efforts to reduce CVD risk in patients with SCZ include pharmaceutical therapies such as metformin and statins (Figure 2). Metformin has been studied as an anti-obesity treatment for antipsychotic-induced weight gain and glucose metabolism dysregulation. Evidence shows that it is also effective in addressing metabolic concerns associated with SCZ [11]. Statins have been studied as a viable option for CVD in SCZ. Several small sample size studies reported beneficial effects of statins for primary prevention of CVD in this psychotic disorder [22,28,29]. An emerging alternative is SSRIs which are under investigation due to their ability to reduce platelet aggregation without increased bleeding risk [30,31]. Recent evidence shows that SSRIs may reduce the risk of coronary disease [13] and decrease the risk of ischemic heart events [32]. In contrast, evidence to support the use of ASA in SCZ is almost non-existent.

In recent years, there has been growing awareness of the role of inflammation in the development of SCZ and its associated symptoms [33]. As a result, ASA and other anti-inflammatory medications have been used as a potential add-on treatment with antipsychotics to foster symptoms resolution. In contrast, ASA’s capacity to reduce CV risk has been ignored. The only study currently available that tangentially relates to this topic is that of Victor et al.

Victor and colleagues performed a retrospective chart review on the utilization of ASA for primary CVD prevention within the psychiatric population. Ninety-three patients aged 50–69 admitted to a state psychiatric hospital between 1 April 2016 to 31 August 2016 who met criteria for ASA use as primary prevention for CVD under USPSTF guidelines were included. Only individuals with serious mental illness such as SCZ and bipolar disorder were considered. Their results, comprising both men and women, showed that out of the 93 patients, 60 were receiving appropriate ASA treatment, with more underutilization of ASA than overutilization according to the 2016 USPSTF guidelines [34]. Underutilization was defined as patients who met the new USPSTF guidelines but were not prescribed ASA, whereas overutilization was defined as patients who did not meet criteria but were still prescribed ASA. Even though this study evaluated the use of ASA in SCZ, its focus was on the application of the new 2016 USPSTF guidelines for those who met criteria within a psychiatric institution. The impact of ASA on CVD in SCZ was not addressed.

## 4. Discussion

CVD is the leading cause of mortality in the U.S. as it accounts for one in four deaths. Current guidelines recommend ASA for patients aged 40–59 with a 10% or greater 10-year CVD risk. Under these circumstances, the benefit of ASA outweighs the potential harm, such as increased risk of bleeding. Given the increased risk of CVD in SCZ, ASA could be especially useful in this patient population.

There are several factors that place individuals with SCZ at higher risk of CVD. This includes a possible proinflammatory state in the brain that plays a role in development of CVD in SCZ [33,35,36,37,38,39,40,41]. Additionally, an inherent predisposition to developing metabolic abnormalities and a sedentary lifestyle also increase the risk of CVD [3,18,42,43]. Thus far, studies have elicited the utility of statins and preliminary evidence indicates SSRIs may be useful under certain conditions [3,22,28,44,45]. However, these approaches come with a host of possible adverse effects such as weight gain, dyslipidemia, myalgia, and liver damage [46,47,48].

It is well established that patients with SCZ have shorter life expectancy than the general population, mostly due to CVD [3,4,24,25]. A study found that individuals with SCZ who did not receive cardioprotective treatment after a myocardial infarction had the highest mortality rate compared with the general population [49]. The study results suggest that some of the increased cardiac mortality among them can be reduced if these patients are efficiently administered combinations of secondary preventive treatments after cardiac events. There have been no studies on the effects of prophylactic cardiac treatment and mortality rates after MI in patients with SCZ compared with the general population [49]. Our review shows that no studies have been done in determining whether primary cardioprotective treatment with ASA can reduce CVD risk in SCZ.

Although the study conducted by Victor et al. addressed ASA utilization for patients with SCZ, it did not directly analyze ASA as a means to reduce CVD risk [34]. In addition, this study was limited to a single psychiatric institution for a short period of time with a relatively small sample size. Inconsistent past medical history documentation could have excluded many patients. This underscores the need for further research on the utilization of ASA for CVD beyond concentrating on USPSTF guidelines, as patients with SCZ are at a higher risk than the general population.

It is important to mention that suicide, together with CVD and cancer, is among the leading causes of mortality in SCZ [50]. Antipsychotics in general [51], and clozapine in particular [52], can reduce the risk of suicide. Regardless of the impact on CVD risk, antipsychotics will continue to be the cornerstone of SCZ treatment. Therefore, it seems imperative to develop a stratified treatment strategy to minimize CVD risk in patients with SCZ and if evidence supports it, include ASA in this equation.

Also of note, is the recently Food and Drug Administration (FDA)-approved medication aimed to mitigate olanzapine-associated weight gain which combines this second generation antipsychotic with samidorphan [53]. While the impact of olanzapine–samidorphan on weight, glucose, and other metabolic parameters is yet to be determined [53], emerging medications like this, as well as naltrexone, which can elicit weight loss in women with SCZ [54], should be part of a stratified treatment approach to minimize CVD risk in SCZ. It is evident that significant research remains to be done, not only about the potential usefulness of ASA in mitigating CVD risk in SCZ, but also about how to personalize and improve the management of metabolic abnormalities often encountered in this patient population.

The strengths of this review include a broad search strategy and comprehensive hand screening process. Nevertheless, there are methodological limitations which must be noted. We excluded papers not available in English. Our literature search may also have been limited if new studies were published after our final review. Finally, our strict screening parameters regarding ASA use in SCZ spectrum disorders may have limited the papers we fully reviewed. As such, important findings may have been missed.

In conclusion, our systematic review points to a significant gap in research on CVD prevention in SCZ and illustrates an obvious need for further work in this area. Although several studies have shown increased CVD risk in SCZ, to date, no research has been conducted on the utilization of CVD preventative treatment such as ASA for SCZ. There is growing evidence of a proinflammatory state and research has shown benefits of ASA utilization for this pathogenic mechanism of SCZ. Adjuvant treatment with ASA could possibly decrease CVD risk and mortality in this high-risk population and therefore, needs to be studied.

## Figures and Tables

**Figure 1 brainsci-13-00368-f001:**
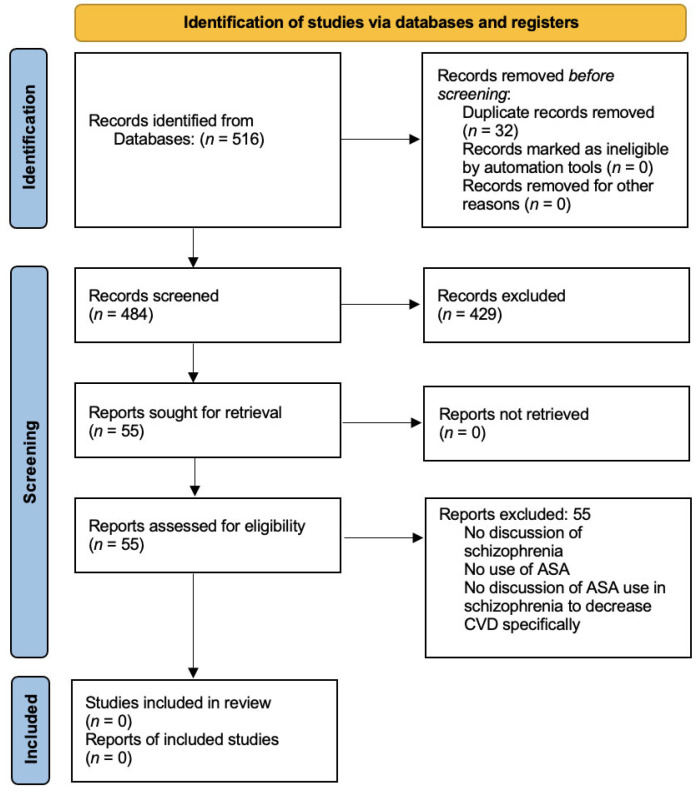
PRISMA 2020 flow diagram for our systematic review including searches of databases and registers.

**Figure 2 brainsci-13-00368-f002:**
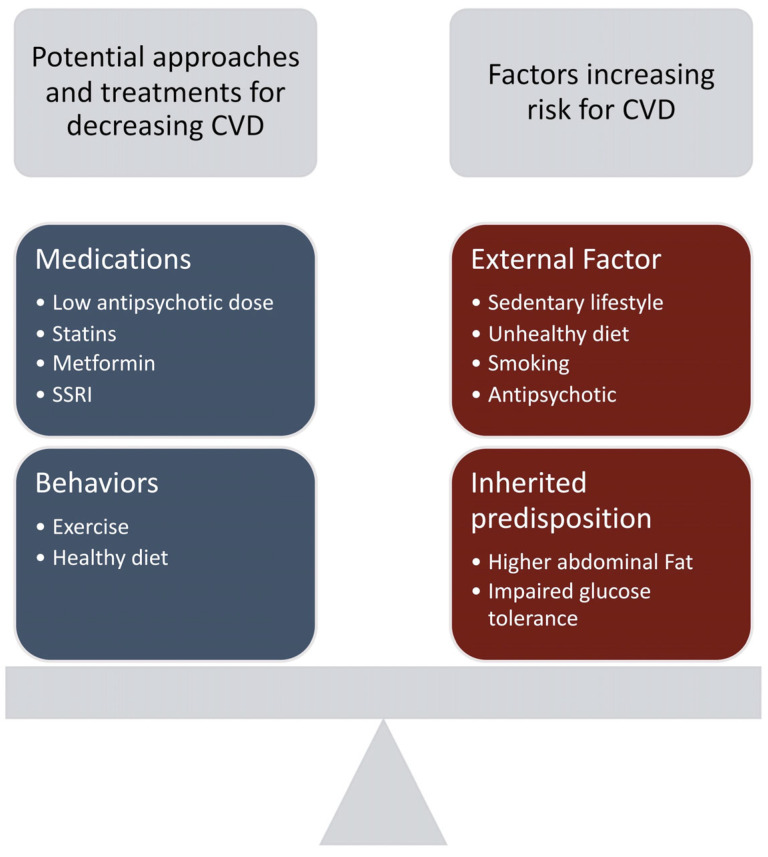
Cardiovascular risk factors and treatment alternatives for patients with schizophrenia (SCZ). The two blocks on the left represent potential approaches to treat and prevent cardiovascular disease (CVD) for patients with SCZ. Blocks on the right represent factors that increase the risk for CVD in patients with SCZ.

## Data Availability

Data are available under request to the corresponding author Alfredo Bellon (abellon@pennstatehealth.psu.edu).

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
