# Peer review of "A Systematic Review on the Potential of Aspirin to Reduce Cardiovascular Risk in Schizophrenia"

_brainsci, 2023, doi:10.3390/brainsci13020368_

Round 1

Reviewer 1 Report

The selected topic is actual for the research in schizophrenia.

However, the statement of the review objectives (line 59-64) did not correspond to the name of the systematic review and to the elicited inclusion and exclusion criteria.

One of the objectives (First, we present evidence on the current understanding of how patients with SCZ are at an increased risk for CDV, line 60-61) is not sustainable with a presented inclusion criteria.

Thus, the result of Section 3.1. (Schizophrenia and cardiovascular risk, line 102) should be removed from the results and moved to the introduction or discussion part.

The search strategy developed with insufficiency of data, for example, the key word was chosen as “antipsychotic agent”, but other synonyms was not taken into account (neuroleptic agent or antipsychotic drug or neuroleptic drug or neuroleptics or antipsychotics or anti-psychotics) what can lead to the missing of the data in the searching.

In the article, the authors frequent duplicate information in three parts: introduction, section 3.1, discussion.

This manuscript does not provide the registration information for the review, including register name and registration number, or state that the review was not registered.

There are not indicate where the review protocol can be accessed, or state that a protocol was not prepared in this paper.

Reviewer 2 Report

Thank you for the opportunity to review the report by Dao and colleagues titled “A systematic review on the potential of aspirin to reduce cardiovascular risk in schizophrenia”. Overall, this is a well-written paper, with a concisely focused topic despite its wide reaching breadth. The aim of the report is highly impactful and timely, and clearly addresses a gap in the field with desperate need for further attention across clinical and research arenas. There are several notable areas with room for improvements that would strengthen the enthusiasm for this manuscript.

Introduction.

-          Please check for typos: CDV instead of CVD in final paragraph of intro

-          Overall, the intro is well developed. One minor suggestion would be to note briefly if there are any conditions, like SCZ, where CVD risk is increased compared to the general population and a specific guideline is provided to give more context to the reader.

Materials and Methods.

-          It should be noted whether or not overlapping samples were used amongst studies included, or if this was accounted for specifically, which would pose a significant confounder if not.

Results.

-          A tabulated clinical demographics / patient characteristics would be helpful to bolster the rigor of the report. Similarly, a tabulated display of included report information may add to the overall impact of the paper for the reader to better grasp the breadth of the reviewed literature. An annotated bibliography of sorts could also be helpful as a supplemental if able to allow for future researchers with similar interests to grow this hugely important topic.

Discussion.

-          Although CVD is increased in risk, so are fatal consequences of undertreated SCZ, like suicide. In light of this, it would be an interesting and important point for the authors to discuss how it could be thought of together. In other words, antipsychotic usage carries the side effects that increase CVD indirectly and directly, but also mitigate symptom burden in SCZ. How would the authors propose to weigh the use of ASA in SCZ population differently than it is weighted in the general population (i.e. it is not the same risk/benefit ratio in the general population compared to SCZ given the different comorbidities and outcomes if left untreated, etc). Similarly in this vein, studies that do aim to study ASA use with concomitant antipsychotic use should have more forgiving guidelines perhaps, given the added risks, and added benefits, from antipsychotics themselves.

-          With the use of olanzapine and samidorphan, a model may be had by which important questions posed in the hypothesis of this report that can be asked about metabolic CVD risks and mitigation attempts in longitudinal ways. Authors may take an opportunity to suggest such studies if there are ways to link it within the scope of their discussion and goals.

Reviewer 3 Report

Dear Editor,
I really appreciate the opportunity to review the manuscript brainsci-2060311 entitled:
"A Systematic Review on the Potential of Aspirin to Reduce Cardiovascular Risk in Schizophrenia"

I commend the authors for describing this critical and timely issue. The paper is interesting and well-written; however, I would like to highlight some issues that merit revision:

The text in the "Results" section reads "All 43 articles were excluded after applying the complete inclusion and exclusion criteria." Next, they begin with results, which are very well written, but it is not clear to the reader how they were obtained, although on page 6 you read "Our review shows that no studies have been done.,." I ask the authors to include a short sentence before the statement of the results where something is indicated indicating what and how it was analyzed.
